# Shedding Light on the Pathogenesis of Splanchnic Vein Thrombosis

**DOI:** 10.3390/ijms24032262

**Published:** 2023-01-23

**Authors:** Sofia Camerlo, Jacopo Ligato, Giorgio Rosati, Giovanna Carrà, Isabella Russo, Marco De Gobbi, Alessandro Morotti

**Affiliations:** Department of Clinical and Biological Sciences, University of Turin, Regione Gonzole 10, Orbassano, 10043 Turin, Italy

**Keywords:** splanchnic vein thrombosis, clonal hematopoiesis, myeloproliferative disorders, hemostasis, cirrhosis

## Abstract

Splanchnic vein thrombosis is a rare but potentially life-threatening manifestation of venous thromboembolism, with challenging implications both at the pathological and therapeutic level. It is frequently associated with liver cirrhosis, but it could also be provoked by myeloproliferative disorders, cancer of various gastroenterological origin, abdominal infections and thrombophilia. A portion of splanchnic vein thrombosis is still classified as idiopathic. Here, we review the mechanisms of splanchnic vein thrombosis, including new insights on the role of clonal hematopoiesis in idiopathic SVT pathogenesis, with important implications from the therapeutic standpoint.

## 1. Introduction

Splanchnic vein thrombosis (SVT) includes portal (portal vein thrombosis, PVT), mesenteric (mesenteric vein thrombosis, MVT) and splenic vein thrombosis, and Budd-Chiari syndrome (BCS) [1]. SVT is generally classified as secondary to an identified risk factor or primitive, unprovoked, when causative factors cannot be identified. While much rarer than common venous thromboembolism (VTE), SVT is often challenging to clinicians for both the identification of the causal disorder and for its therapeutic management. It is worth noting that SVT is actually a heterogeneous disease for many reasons [2]. SVT involves different veins that originate from different organs, implying specific clinical presentations and the risk of organ failure. SVT can be associated with different pathological conditions, therefore requiring the treatment of the underling condition together with anticoagulation. Lastly, SVT could be associated with variable aberrations of the hemostatic balance, that can also be associated with a potential increase in bleeding risk, as observed in patients with cirrhosis or esophageal varices due to portal hypertension. Historically, SVT has been described following the identification of subgroups, according to the anatomical distribution of the thrombosis (BCS, PVT, SVT, MVT) or the underlying conditions (cirrhotic, cancer, myeloproliferative neoplasia, abdominal infections or idiopathic). In the era of molecular medicine, new investigations are mandatory to better dissect the physiopathology of SVT, in particular idiopathic SVT, which remains a frequent diagnosis. Providing a more precise diagnosis may favor a better stratification of patients and a better therapeutic approach. In this review, we will discuss the known pathogenic mechanisms of SVT and we will comment on the SVT therapeutic implications. Lastly, we will provide new insights on the role of clonal hematopoiesis in SVT pathogenesis.

## 2. Results

### 2.1. Epidemiology

SVT is an unusual manifestation of venous thromboembolism, with an overall incidence rate at least 25 times lower than those observed in usual sites [3]. The most common site of SVT is represented by portal vein thrombosis (PVT), followed by mesenteric vein thrombosis (MVT) (incidence 0.7–2.7/100.000), with BCS as the least frequent (incidence 0.5–1/1.000.000) [4,5,6]. In almost 40% of cases, thrombosis in multiple sites have been observed at diagnosis [7]. Notably, the incidence of SVT is highly variable among age groups and populations, although it appears more frequent in male.

### 2.2. Risks Factors and Correlations

SVT is frequently associated with liver cirrhosis and solid cancers, which represent half of the cases. Other risk factors include myeloproliferative neoplasm (MPN), thrombophilia, abdominal surgery, abdominal infectious and inflammatory disorders [3,4,5,6,8]. However, unprovoked SVT still represents 15% to 27% of all SVTs [9].

### 2.3. Cirrhosis

Liver cirrhosis corresponds to the final stage of several chronic liver diseases and is characterized by a remodeling of the liver architecture, caused by the replacement of healthy liver parenchyma with fibrotic tissue, thus leading to the setting of regenerative nodules, fibrous septa and portal hypertension [10]. The causes can vary between vascular, autoimmune, exotoxin, metabolic and vital infections [11]. Splanchnic vein thrombosis (SVT), particularly portal vein thrombosis (PVT), is the most common complication in patients with liver cirrhosis [11,12,13,14], even if mesenteric veins thrombosis can also be observed. It seems that this relationship is due to an imbalance in the production of thrombin [15] and blood stasis because of portal hypertension (particularly in case of PVT) [15,16,17]. Interestingly, thrombosis distribution and liver disease severity (measured through the Child-Pugh classification) did not correlate each other [18].

Cirrhosis-associated SVT is a distinct entity of splanchnic thrombosis (Figure 1) due to the association with a specific aberrant hemostatic impairment. Patients with cirrhosis display various modifications in the clotting system, as well as in the number and function of platelets. Moreover, patients with cirrhosis could present with portal hypertension and the presence of esophageal varices, which can further render complex the therapeutical approach, with a potential increase in the bleeding risk. Patients with cirrhosis-associated SVT require a specific diagnostic and therapeutical approach, reviewed elsewhere [19,20,21]. It is worth noting that the attempt to resolve SVT in cirrhosis is associated with the reduction in deaths.

### 2.4. Thrombophilia

Inherited and acquired thrombophilia correlate with an increased risk of thrombosis in unusual sites, such as cerebral vein thrombosis or splanchnic vein thrombosis, more frequently than in usual sites [8]. Mainly, factor V Leiden (rs6025 or F5 p.R506Q) and, to a lesser extent, prothrombin G20210A polymorphism, are important risk factors for BCS and PVT. On the contrary, deficiencies of antithrombin, protein C and protein S are infrequently diagnosed [22,23]. In regard to the antiphospholipid syndrome, anti-cardiolipin antibodies (aCL) are the antiphospholipid antibodies most frequently associated with the development of SVT; in particular, IgG aCL can be identified in patients with Budd Chiari Syndrome and PVT [24,25]. As a consequence, the guidelines recommend screening for antiphospholipid antibodies in patients with new diagnosis of idiopathic SVT [26].

### 2.5. Autoimmune Disorders

Autoimmune diseases are less common risk factors of SVT. In particular, Behçet’s disease is associated with an increased risk of thrombosis. The most common type is lower limb venous thrombosis, but it is also associated with unusual site thrombosis, such as superior and inferior vena cava syndromes, Budd-Chiari syndrome and cerebral venous thrombosis [3,27]. Inflammatory bowel disease (IBD), which includes Crohn’s disease (CD) and ulcerative colitis (UC), is also associated with an increased VTE risk of about 2–3 times compared to the general population, with no difference between CD and UC. In particular, SVT is a rare complication of IBD, and studies have demonstrated that its prevalence in this population is 0.4–1.45% [28]. The main risk factors for thrombosis are intra-abdominal surgery, intra-abdominal infection, and flare-up of the disease. The onset of thrombosis worsens the prognosis and increases the mortality in IBD patients. Thus, the pharmacological thromboprophylaxis with LMWH or fondaparinux is recommended in patients with IBD during hospitalization [29].

### 2.6. Cancer-Associated SVT

Cancer patients have a higher associated thrombotic risk than the normal population and neoplastic patients with a concomitant diagnosis of thrombosis exhibit a greater risk of mortality and disease progression [30,31,32]. Splanchnic venous thrombosis (SVT) can be found in several types of solid tumors, particularly those originating from the digestive tract (particularly in hepatic, pancreatic and digestive tract carcinomas). In such cases, PVT is the most frequent thrombotic manifestation [33]. SVT is mainly found in advanced diseases, assuming a role as a marker of occult cancer and prognostic marker in liver and pancreatic tumors [33,34,35]. Similarly, a high correlation between hepatocellular carcinoma (HCC) and PVT has been described [36,37]. From a pathogenetic point of view, patients with HCC develop SVT because of the cirrhotic state, the neoplastic invasion, hyperinflammation state and production of Extracellular Neutrophil Traps (NETs). Furthermore, chemotherapy and other medical/surgical therapies and devices in these patients could further augment the risk of developing SVT. It is also noteworthy to consider that HCC can overproduce TPO and coagulant factors, therefore altering the hemostatic balance [33,38,39,40]. Additionally, it has been found that levels of Annexin 2 (ANXA2) are increased in HCC [41], with a potential role in triggering thrombosis or bleeding [42]. In patients with pancreatic cancers, a direct link to thrombosis has also been observed, due to the release from pancreatic cancer cells of procoagulant factors (in particular, tissue factor and thrombin), therefore increasing the risk of thrombosis [43]. In various gastrointestinal cancers, SVT, and in particular PVT, is frequently diagnosed [33], particularly in those cancers with extensive vascular involvement [44].

### 2.7. Myeloproliferative Neoplasms- Associated SVT

Patients affected by Philadelphia negative MPN (polycythemia vera, essential thrombocythemia, and primary myelofibrosis) present with an increased risk for both arterial and venous thrombosis compared to the general population [45]. Approximately 10% of all SVT are MPN-related. It is estimated that about 5% of patients with PVT and 25–50% of those with BCS are affected by MPN. The diagnosis of MPN can be synchronous to thrombotic manifestations, or it may precede the hematological diagnosis. For this reason, the evaluation for MPN should be performed in non-cirrhotic SVT, regardless of the hematologic manifestations [46,47]. Quantitative and qualitative alterations of platelet, erythrocytes, leukocytes and endothelial cells, caused by gain-of-function mutations in JAK2, CALR or MPL, are implicated in the pathogenesis of thrombosis in MPN [45,48]. It is worth noting that the association of CALR and MPL mutations with SVT is weaker than JAK2 mutations [47]. The prevalence of the JAK2 mutations is higher in SVT than in usual VTE (32.8% v.s. 0.88%), and it is found in almost all patients (94.7%) with MPN and thrombosis in unusual sites [49,50]. In this respect, a recent report has been published on the assessment of JAK2 mutations (including exon 12) by NGS in non-cirrhotic SVT patients [51]. Notably, JAK2 mutations were reported as a frequent finding in SVT patients, further supporting the link between SVT and MPN.

### 2.8. COVID-19 Infection

SARS-CoV-2 (severe acute respiratory syndrome coronavirus 2) is responsible for the systemic disease called coronavirus-2019 disease (COVID-19), characterized by general symptoms, pulmonary conditions (including ARDS) [52] and thromboembolism [53,54]. In addition to pulmonary embolism and deep vein thrombosis [55], cases with thromboses in the splanchnic venous district have also been reported [54,56]. The correlation between SVT and COVID-19 is not age-dependent (involving both young and old people), but a small prevalence of the male sex was observed [54,56,57]. Most frequently, the portal vein has been involved. Different mechanisms have been implicated in the pathogenesis of SVT thrombosis in COVID-19 patients [58], including: high release of proinflammatory cytokines, endothelial direct damage by the virus, increased procoagulant factors (notably VIII-F, vWF and V-F) and NETs [54,59,60]. Furthermore, a role for platelets has also been deciphered in COVID-19-induced thrombosis, as we reviewed previously [61]. The probable life-threatening condition caused by SVT-COVID19 could be catastrophic when it occurs in a pre-hypercoagulable state [62]. On the other hand, vaccination with adenoviral vector-based COVID-19 may also cause thrombocytopenia and SVT: this condition, known as VITT (vaccine-induced immune thrombotic thrombocytopenia), is due to the interaction between platelet activating antibodies and platelet factor 4 [63].

## 3. Clinical Manifestations of SVT

Clinical manifestations of SVT are very heterogeneous [64]: abdominal pain is the most common, followed by gastrointestinal (GI) bleeding and ascites. Other GI symptoms, such as nausea, vomiting, changes in bowel habits and systemic symptoms (e.g., anorexia and fever), are also frequent [65]. However, approximately one-third of cases are asymptomatic, and the diagnosis is incidentally detected at abdominal imaging tests. Each site of thrombosis can occur with specific symptoms and the sudden onset of abdominal pain is used to distinguish acute and chronic SVT. Chronic PVT typically presents signs of portal hypertension, such as hypersplenism, ascites, esophageal varices, and the presence of portal cavernoma or other portosystemic collateral veins [66]. Intestinal infarction is a complication of acute and subacute MVT. Abdominal pain, ascites and hepatomegaly are a characteristic triad of symptoms of BCS, and acute liver failure can be a manifestation of fulminant BCS [67].

### Clinical Evolution of SVT

The prognosis of SVT depends on the underlying cause: active cancer, MPN and liver cirrhosis [68]. The highest and the lowest mortality rates are reported, respectively, in patients with solid cancer and unprovoked SVT [69]. An international prospective cohort study [4] and a Danish population-based cohort study [70] have evaluated the short and long-term prognosis of SVT. SVT is associated with an increased risk of major bleeding and thrombotic events compared to the general population, and the rate of adverse events is particularly relevant in the first 30 days [70]. Gastrointestinal bleeding is the most frequent hemorrhagic manifestation and the main recurrent thrombotic events occurred in the splanchnic veins or as VTE in other usual sites. Liver cirrhosis represents the most important risk factor for both major bleeding and thrombotic events, whereas, in non-cirrhotic patients’ myeloproliferative neoplasms, solid cancer, unprovoked SVT and the male sex are predictors of recurrence, and the risk of major bleeding appears to be similar to patients with usual site VTE. The overall incidence of thrombotic events is approximately double that of major bleedings, but a prolonged duration of anticoagulant treatment is associated with a significant reduction in both risks [71].

## 4. Diagnostic Considerations

### 4.1. Imaging Tests for the Diagnosis of SVT

The diagnosis of SVT is based exclusively on the use of imaging tools [72]. In particular, angiography was the gold standard, recently replaced by doppler-ultrasound (DUS), computed tomography (CT) and magnetic resonance (MR) [73,74]. DUS is the first line diagnostic test for PVT and BCS, with a sensitivity of 89–93% and a specificity of 92–99%; moreover, it can also provide insights on the extension of the thrombosis, with the detection of partial or complete obstruction of the portal and intrahepatic veins. Abdominal CT or MR are required to confirm the presence and the extent of PVT and BCS, with a sensitivity of 90% and a specificity of 99%, when DUS is not certainly diagnostic for thrombosis. Regarding the diagnosis of MVT and splenic vein thrombosis, CT and MR are the reference imaging tools, with sensitivity of 91–95% and specificity of 94–100% for CT, and sensitivity and specificity of approximately 100% with MR [72]. DUS has a good specificity (100%) but low sensitivity (70–90%) due to bowel gas, which can interfere with a good acoustic window and the presence of several collateral veins near the splenic hilum, which can be confused with the splenic vein [75,76]. The use of liver biopsy is reserved to confirm rare forms of BCS involving only small intrahepatic veins or to exclude other hepatic disorders, such as veno-occlusive disease [74].

### 4.2. The Role of Blood d-Dimers Test in the Diagnosis of SVT

The role of the D-dimer test is controversial for the diagnosis of SVT due to the high false positive rate; the sensitivity of this laboratory test is 96%, but its specificity is 25% [77]. Indeed, the D-dimer can be increased in several other conditions, such as liver cirrhosis, hepatocellular carcinoma or other gastrointestinal tumors and inflammatory bowel conditions, regardless of the presence of thrombosis. Therefore, the D-dimer test is not required to confirm or rule out the diagnosis of SVT [78,79].

## 5. Therapeutic Implications

The therapeutic approach of SVT is a clinical challenge and considers the manifestations and the site of thrombosis, the risk of SVT progression, recurrence and bleeding [80]. The decision regarding when to start, as well as the type and the duration of anticoagulant therapy is often made empirically [81]. First, while the ACCP guidelines suggested no anticoagulation for asymptomatic SVT [82], the 2020 ISTH guidance advices that the same therapeutic approach should be followed for patients with symptomatic and incidentally SVT, as the risk of recurrent VTE is similar [83]. In regard to the beginning of treatment, in the absence of absolute contraindications, anticoagulation should be started early (within first two weeks) to improve the recanalization rates and reduce the risk of complications, such as thrombosis progression, intestinal infarction or ischemia and incidence of splanchnic hypertension [81]. However, before anticoagulation, clinicians should consider variceal screening, performing esophagogastroduodenoscopy in patients with signs of portal hypertension. Indeed, medical prophylaxis with beta-blockers and endoscopic treatment with variceal band ligation reduce the incidence of bleeding. Moreover, considering patients with shock, high lactate levels or signs of peritonitis, perforation or acute major gastrointestinal bleeding, the ISHT guidelines recommend immediate surgical treatment or invasive procedure (such as systemic or catheter-directed thrombolysis) [84,85]. UFH or LMWH, eventually followed by VKAs (INR range 2–3), is recommended as an anticoagulation for unusual site VTE [81]. The optimal duration of anticoagulant treatment for splanchnic vein thrombosis remains unclear. The guidelines recommend at least 3–6 months of anticoagulant treatment [73,82]. However, in patients with unprovoked SVT or a permanent prothrombotic condition, an indefinite anticoagulant treatment duration is advocated to prevent VTE recurrence [85]. A different approach is required by BCS. Due to its severity, interventional procedures, such as percutaneous angioplasty, systemic thrombolysis, or TIPS, up to liver transplantation can be necessary. In regard to medical therapy, anticoagulation is required, indefinitely [85,86].

### 5.1. A Role for Direct Oral Anticoagulants (DOACs)

The use of DOACs, specific inhibitors of factor Xa or factor IIa, for the treatment of SVT remains contentious [87]. This is due to the registration trials for DOACs only including patients with PE or DVT: the studies do not include patients with any type of SVT, despite their advantages over anti-vitamin Kappa anticoagulants (VKAs) being relevant for patients with unusual site VTE [88,89,90]. The 2020 ISTH guidelines suggested the possibility of using DOACs in patients without cirrhosis, with acute symptomatic SVT [85]. Retrospective studies and limited prospective data have demonstrated the safety and effectiveness of DOACs compared to VKAs and low molecular weight heparin (LMWH) in this cohort of patients [28,87,91,92]. In a retrospective trial, Naymagon et al. compared the use of DOAC (Apixaban, Rivaroxaban and Dabigatran) with LMWH and Warfarin for the treatment of portal vein thrombosis in patients without cirrhosis. The rate of thrombosis’ resolution was higher in patients treated with DOAC than with LMWH of warfarin, and the rate of major bleeding was reduced with the DOAC compared with warfarin [91]. The same authors also analyzed a cohort of patients with PVT and inflammatory bowel disease and similar results were reported concerning the safety and effectiveness of DOACs [28]. A recent study conducted by Ageno et al. evaluated the use of Rivaroxaban to treat portal, mesenteric and splenic vein thrombosis. The RIVA-SVT 100 study is an international, single group assignment, open-label, prospective cohort study; it excluded patients with cirrhosis but included patients with solid cancer, hematologic malignancies and unprovoked SVT. The study showed that the use of rivaroxaban, compared to heparin and VKAs, is safer in terms of bleeding risk; it is also effective considering the recanalization of splanchnic veins at 3 months [92]. Limited data are available concerning the efficacy and safety of the use of DOACs for secondary prophylaxis of VTE. The prospective study conducted by Serrao et al. compared the use of DOACs with warfarin in the chronic treatment of SVT. There was no difference in the thrombotic events and bleeding rate between the groups. This study suggested that DOACs could represent an effective and safe alternative to warfarin for secondary prophylaxis in SVT patients at high risk of the recurrence of thrombosis [93].

### 5.2. Cytoreductive Therapy in MPN-Associated SVTs

In addition to anticoagulation, cytoreductive therapy with hydroxyurea, peg-interferon alfa or the JAK2 inhibitor, ruxolitinib, is offered to all patients with MPN who develop an SVT to reduce the risk of recurrent thrombosis [94]. However, the efficacy of cytoreductive therapy in reducing SVT recurrence, beyond the use of anticoagulation, is not well established. The management of SVT in patients with isolated JAK2 mutations or morphological MPN diagnosis with normal blood counts is debated. In view of the unclear benefits, the cytoreductive therapy is not administered to all patients [46,95].

### 5.3. New Insights in SVT: A Role for Clonal Hematopoiesis?

Recently, we have reported the increased incidence of clonal hematopoiesis of indeterminate potential (CHIP) in a cohort of idiopathic-SVT [96]. Clonal hematopoieisis (CH) defines a population of hematopoietic cells with one or more somatic mutations or copy number alterations, able to expand overtime with a positive selection pressure. The term CHIP refers to the presence of somatic mutations in leukemia-associated driver genes with a variant allele frequency of more than 2%, in the absence of any hematological cancer [97]. The detection of somatic mutations was reported as a rare condition in young individuals, with an increase above 70 years of age [98]. In this report, patients with CH were associated with an increased risk of mortality for all causes, including cancer development. Interestingly, CHIP has also been associated with an increased risk of cardiovascular disease and, in particular, atherosclerosis [99]. Notably, the modeling of CH in mice was also associated with an increased development of vascular lesions [99], therefore pointing to a potential role for CH in the development of thrombosis. In this respect, CH was shown to be associated with unprovoked pulmonary embolism, suggesting that CH should be considered as a novel risk factor for thromboembolism [100].

While searching for mutations with a next generation sequencing (NGS) approach with MPN associated genes (ABL1, ASXL1, BRAF, CALR, CBL, CEBPA, CSF3R, DNMT3A, ETV6, EZH2, FLT3, HRAS, IDH1, IDH2, JAK2, KIT, KRAS, MPL, NPM1, NRAS, PTPN11, RUNX1, SETBP1, SF3B1, SRSF2, TET2, TP53, U2AF1, WT1, ZRSR2), we discovered that almost 50% of patients with idiopathic-SVT display somatic mutations. Interestingly, we identified the most frequently mutated gene in DNMT3A. While waiting to be confirmed on larger cohort of patients, our data point to a potential role of specific mutations in increasing the risk of SVT. DNMT3A mutations, together with the TET2 and ASXL1 genes, have the ability to regulate the effect of DNA methylation and, therefore, to modulate gene expression. Interestingly, mutations in the DNMT3A, TET2 or ASXL1 genes have already been reported to increase the thrombotic risk of patients with polycythemia vera. In particular, the presence of at least one mutation in the TET2, DNMT3A and ASXL1 gene is associated with the increase in risk of vascular events by six folds [101]. It could be speculated that mutations in DNMT3A could modulate the expression of various genes involved in the inflammation response, therefore favoring the development of thrombosis. Further investigations are mandatory to both confirm the pathogenetic role of CHIP in the development of SVT and to assess the pathogenetic mechanism linked to specific mutations.

It is, however, tempting to assume that these results suggest that CHIP should be assessed in the group of idiopathic SVT, offering new insights in the management of these patients. Similar to our observations, other groups have proposed to screen patients with idiopathic-SVT using NGS techniques, allowing the identification of recurrent mutations, such as JAK2-ex12 mutations [51]. Patients with CHIP-associated SVT should indeed represent a novel entity and should require a tighter control over time, due to the risk of progression into a MPN disorder. These patients should also probably require a prolonged anticoagulation due to the potential risk of thrombotic relapse, as observed in MPN patients.

## 6. Conclusions and Future Directions

While rare as a clinical entity, the identification of a patient with SVT is still challenging, requiring an individualized diagnostic and therapeutical approach. As we report in Figure 1, all SVT patients must be classified as secondary to cirrhosis or cirrhosis independent. In the first scenario, cirrhosis-associated SVT should be treated as an independent entity, where thrombosis and the risk of bleeding should be tightly monitored and an appropriate therapeutic approach should be considered. In cirrhosis-associated SVT, it is indeed important to consider how liver disease can affect the hemostatic balance, as well as the frequent and concomitant thrombocytopenia. In the other cases, beside patients with abdominal surgery and/or abdominal infections, it is mandatory to rule out the presence of gastrointestinal cancers and myeloproliferative disorders, which are the most common cause of SVT in this subgroup of patients.

While idiopathic SVT was considered to account for about 25% of the cases, we recently showed that a portion of idiopathic SVT patients presented with CHIP, therefore identifying a novel entity: CHIP-associated SVT. This type of SVT should require a tighter follow up time due to the increased risk of progression into a myeloproliferative disorder and a potential prolonged/indefinite anticoagulation for the risk of thrombosis relapse. 

## Figures and Tables

**Figure 1 ijms-24-02262-f001:**
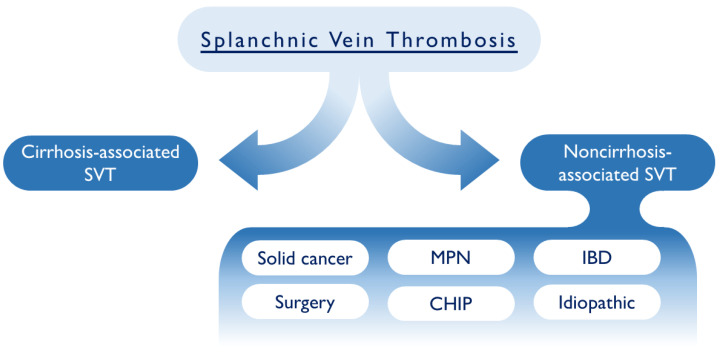
Schematic representation of SVT classification. MPN: myeloproliferative neoplasms, IBD: inflammatory bowel disease; CHIP: clonal hematopoiesis of indeterminate potential.

## Data Availability

Not applicable.

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
