# Peer review of "Shedding Light on the Pathogenesis of Splanchnic Vein Thrombosis"

_ijms, 2023, doi:10.3390/ijms24032262_

Round 1
Reviewer 1 Report
Dear authors,
The topic of the study is interesting but I was disappointed by the lack of care in academic writing both minor (many acronyms have no explanation, journal reference style not used, recent systematic reviews not quoted, Figure without description of abbreviations etc.) and major: not a single reference in two consecutive paragraphs quoting other scientists' work (lines 138-164) is a classic definition of plagiarism, and as such disqualifies your work.
Author Response
We agree with the comments of this reviewers. We apologize for the lack of citation of various works that we have erroneously removed from the submission. We are deeply sorry for this mistake, that has been corrected as described below.
Q1 many acronyms have no explanation
R1: all acronyms have been explained in the manucript
Q2 journal reference style not used
R2 journal reference style has been corrected
Q3 recent systematic reviews not quoted
R3 we have significantly increase the number of reviews including important and recent ones.
Q4 Figure without description of abbreviations
R4: abbreviations have been described in the legend.
Q5 not a single reference in two consecutive paragraphs quoting other scientists' work (lines 138-164)
R5 we apologize for this inconvenience. References have been included.
Reviewer 2 Report
Hallo,
My comments are as follow:
1. Line 143. Authors tell about abdominal imaging tests. So, in this context please provide the information about sensitivity and specificity of ultrasound (US) , CT -scan and MRI procedures in diagnosis of SVT - you can present this problem in the larger context of portal thrombosis imaging.
2. Please discuss the role of blood d-dimers test in the diagnosis of SVT.
Author Response
We thank reviewer for his/her comments that helped us to improve our manuscript.
Questions
Q1. Line 143. Authors tell about abdominal imaging tests. So, in this context please provide the information about sensitivity and specificity of ultrasound (US) , CT -scan and MRI procedures in diagnosis of SVT - you can present this problem in the larger context of portal thrombosis imaging.
R1: we thank reviewer for this comment. A novel chapter has been included in this review. Please see from line 190.
Q2. Please discuss the role of blood d-dimers test in the diagnosis of SVT.
R2: we agree with this comment. A novel chapter has been included. Please see from line 208.
Round 2
Reviewer 1 Report
The authors have improved the text significantly and addressed all of my comments. I hope that the lack of references mentioned in the first review was an honest mistake while editing...